

# Quantifying the effect of low-frequency fatigue dynamics on offshore wind turbine foundations: a comparative study

Negin Sadeghi[1], Pietro D'Antuono[1], Nymfa Noppe[1,2], Koen Robbelein[1,2], Wout Weijtjens[1], Christof Devriendt[1]

[1]Vrije Universiteit Brussel (VUB), OWI-Lab, AVRG, Brussels, 1050, Belgium
[2]24SEA, Brussels, 1000, Belgium

*Correspondence to*: Negin Sadeghi (negin.sadeghi@vub.be)

**Abstract.** Offshore wind turbine support structures are fatigue-driven designs subjected to a wide variety of cyclic loads from wind, waves, and turbine controls. While most wind turbine loads and metocean data are collected at short-term 10-minute intervals, some of the largest fatigue cycles have periods over one day. Therefore, these low-frequency fatigue dynamics (LFFD) are not fully considered when working with the industry-standard short-term window. To recover these LFFDs in the state-of-the-industry practices, the authors implemented a short-to-long-term factor applied to the accumulated short-term damages, while maintaining the ability to work with the 10-minute data. In the current work, we study the LFFD impact on the damage from the Fore-Aft and Side-Side bending moments and the sensors' strain measurements and their variability within and across wind farms. For an S-N curve slope of m=5, up to 65 % of damage is directly related to LFFD.

## 1 Introduction

In the next decade, several European offshore wind farms will start approaching their end of as-designed service life (Archer et al., 2014). As a result, the industry must prepare for making complex decisions such as lifetime extension, repowering, optimizing, or decommissioning (Pakenham et al., 2021). A viable option to support this decision would be using structural health monitoring (SHM), through which it is possible to update the as-designed lifetime figures with in-situ data. To this aim, strain sensors are installed at the tow-er/transition piece interface to measure the strains, collecting the support structure's strain over an extended period of up to several years. The strain data, as well as other sources of operational and environmental data, is commonly stored as segmented datasets in 10-minute blocks (short-term) to be consistent with the duration of design load cases (DLCs) time series defined and used in the design phase (DNVGL-ST-0437, 2016). Time series are commonly cycle-counted into a histogram through standardised algorithms (ASTM, 2017; ASTM E08, 2005), among which rainflow (Amzallag et al., 1994; Socie, 1992) is the most commonly used. The cycle-count histogram summarises the mean-range (or mean-amplitude) couples and their cycle counts (i.e., the frequency of occurrence within the considered time window).

One cycle represents a closed hysteresis loop in the time series, while a half cycle, also known as residual, is part of a hysteresis loop that did not have the time to complete within the time window. When analysing signals, it is important to



consider the length of the time window used for analysis. Ideally, an infinitely long signal would not have any residuals. However, when working with finite-length signals, residuals can appear with a frequency of at least 1 divided by the time window length (Sutherland, 1999). Therefore, in the industrial approaches, if we analyse multiple 10-minute-long signals separately, we may obtain different results compared to analysing a concatenated signal that spans a longer period. This is

because the longer time span may reduce the impact of residuals on the analysis. This poses a challenge when considering the low-frequency fatigue dynamics (LFFD) of a (offshore) wind turbine primarily caused by the slow variations in wind speed and direction which have periods well beyond the default 10-minute window.

In this paper, we are interested in the additional damage that might remain unnoticed if the LFFD impact was not taken into account in common practices, such as the commonly used recommended practice (DNVGL-RP-C203, 2019) for the

structural design of offshore wind turbines. Our research intends to provide practical industrial applications, so it must follow the most frequently used industry norms for predicting fatigue life times.

When processing 10-minute-long signals, these LFFD cannot be captured and any possible fatigue damage calculation rule (e.g. linear Palmgren-Miner (Ciavarella et al., 2018; Miner, 1945; Palmgren, 1924) or non-linear models (Hectors and De Waele, 2021)) will result in possibly non-conservative fatigue life estimates as some of the largest cycles remain

unaccounted for. On the other hand, concatenating the segmented signals and cycle-counting the output is of very little practical applicability, given that there could be multiple years of SHM data sampled at frequencies well above 1 Hz (D'Antuono et al., 2022) and results can no longer be matched with the 10-minute design load cases (DLCs). To consider the LFFD effect, some researchers adopted various strategies.

An idea is to count half cycles as full cycles, but this would not cover the whole effect of the LFFD. In fact, counting

residuals as full cycles is not a common practice as it does not have any physical rationale, as there is no hysteretic loop being artificially closed. Based on IEC standard (IEC-61400-13, 2001) it is recommended to treat half cycles as half.

Because of the potential discrepancy, some research on the role of LFFD in wind energy has been conducted. On a small wind turbine (Micon M150), Larsen and Thomsen (1996) established and used a framework for an approximative treatment of low-frequency contributions. They demonstrated that the low-frequency part of the blade flap-wise moment accounts for

roughly 8 %/1 % for large/small Basquin slopes in terms of equivalent moments. For large/small Basquin slopes, the low-frequency portion contribution to the corresponding moment for the rotor tilt is roughly 2 %/0.4 %. When considering the LFFD effect of a Vestas V90-2.0 MW wind turbine, Pacheco et al. (2022) reported that the damage calculated with a tri-linear S-N curve from Euro-code 3 and from a 24h time series was 11 % higher than the damage obtained with a 10-minute time series.

The present paper adopts an algorithm that allows recovering the LFFD through the sequences of residuals of the segmented data and their cycle-count histograms. The algorithm was first proposed by Amzallag et al. (1994), then Marsh et al. (2016) applied it specifically to wind energy on a multi-megawatt offshore wind turbine. For m=3 and m=5, respectively, Marsh et al (2016) reported an increase of up to 10 % and 170 % in the final damage when LFFD was accounted for. Finally, Sadeghi et al. (2022) validated the procedure using three years of SHM data coming from an offshore wind turbine monopile



foundation. In that work, the LFFD recovery algorithm has been used to calculate an LFFD-factor that, once applied to the sum of the damages from each short-term dataset, allows the recovery of the detrimental effect of those low-frequency/high-range fatigue cycles.

The LFFD-factor is equal to the ratio of the damage calculated using long-term data to the damage calculated using short-term data. Both Marsh et al (2016) and Sadeghi et al. (2022) have found that said factor converges to a fixed value after a specific amount of time, suggesting that once the LFFD-factor is converged there is no need to continue to work with long-term cycle counting. Moreover, the LFFD-factor provides a direct insight into the relative importance of LFFD on overall fatigue.

The LFFD-factor will play a role in assessing the fatigue life of operational OWTs to account for the long-term cycles to which the support structure is subjected and meanwhile collect data in 10-minute windows, far better suited for comparing with DLCs or fatigue prognoses (Noppe et al., 2020). However, Marsh et al. (2016) already showed that the LFFD-factor is highly dependent on the gradient of the chosen S-N curve. Moreover, the LFFD-factor presumably depends on the stress history, which is site, turbine, and even sensor specific, as the amount and size of LFFD cycles will depend on the prevailing environmental and operational conditions. This work investigates the behaviour of the LFFD-factor under various scenarios using SHM data from several OWTs.

## 2 Measurement setup and methodology

OWI-lab had access to data of four OWTs across two wind farms which were geographically close to each other, with near identical metocean conditions. The first farm includes one of the first generations of 3MW OWTs, while the second farm has a larger and newer generation of 9MW OWTs, installed in deeper waters and with larger rotor diameters compared to the first wind turbine in Farm 1. While both farms used monopile foundations, the larger monopiles in Farm 2 have a far larger impact from wave-loading-induced fatigue compared to the smaller monopiles at Farm 1, as the larger wind turbines have lower natural frequencies, closer to the wave spectra (Laszlo et al., 2016). We have three years of data for one turbine (Farm 1), while for the remaining three wind turbines (Farm 2), we have collected one year of data. As shown in Fig.1, each turbine is equipped with six longitudinal strain gauges mounted 60° apart and installed at the tower-transition piece interface, 16/19m above the LAT (Lowest Astronomical Tide). The three turbines of Farm2 (F2) have similar models and structural designs. The heading (position) of strain sensors along the circumference of the tower is almost similar for all four turbines. Also, the dominant wind direction which was almost similar for the four turbines is shown in this figure. Sensors S2 and S4 of Turbine1 and S6 of Turbine3 were faulty and were neither used to calculate bending moments nor to calculate the heading-specific damage.



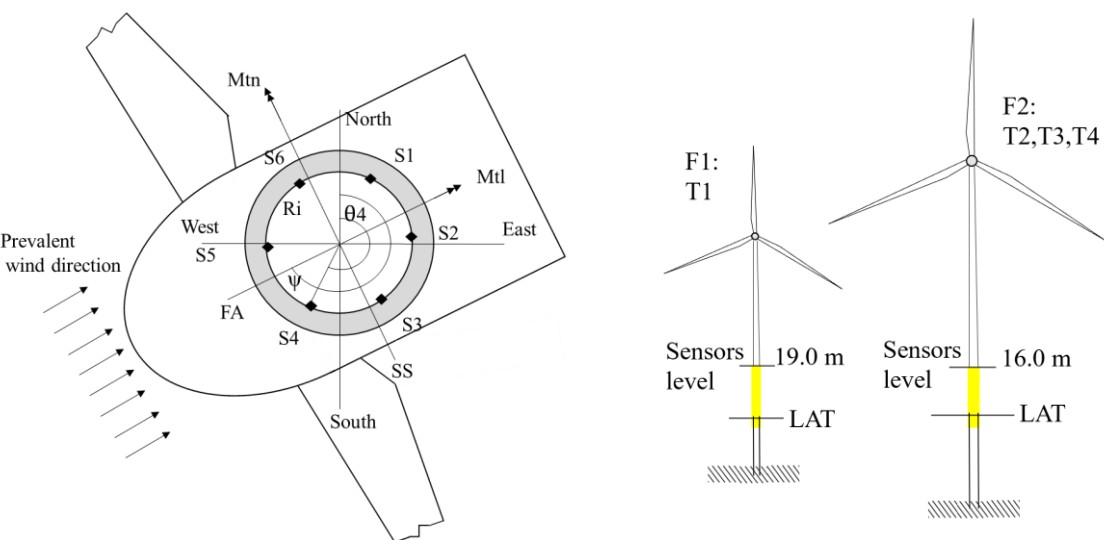

**Figure 1: Schematics of the instrumentation setup for Farm1 and Farm2. Sensors are installed at the Tower-TP interface level.**

From the strain measurements at the sensors (S1 to S6), we can calculate stresses (σ) and bending moments (M) at any location in the structure through Hooke's law, Eq. (1), Navier's formula for axial stress, Eq. (2) and a rotation matrix, Eq. (3):

$$\sigma_{zz,j} = E\ \varepsilon_{zz,j}, \tag{1}$$

$$\sigma_{zz,j} = \frac{F_N}{A} + R_i \cdot \left[ \frac{M_{North-South}}{I_C} \cdot \sin(\theta_j) - \frac{M_{East-West}}{I_C} \cdot \cos(\theta_j) \right], \tag{2}$$

$$\begin{Bmatrix} M_{tl} \\ M_{tn} \end{Bmatrix} = \begin{bmatrix} \cos(-\psi + \pi) & \sin(-\psi + \pi) \\ -\sin(-\psi + \pi) & \cos(-\psi + \pi) \end{bmatrix} \cdot \begin{Bmatrix} M_{North-South} \\ M_{East-West} \end{Bmatrix}, \tag{3}$$

Where E is Young's modulus, $\varepsilon_{zz,j}, \sigma_{zz,j}$ are respectively the axial strain and stress at the j-th sensor, $F_N$ is the normal load, $\theta_j$ is the clockwise angle between the North-South axis and the j-th sensor, $R_i$ is the inner radius of the sensor location, A is its cross-sectional area, $I_C$ is its area moment of inertia, and ψ can be any desired heading. In Eq. (3), ψ is selected as the average yaw angle within a 10-minute time window, so the rotation matrix will return the Fore-Aft (FA) and Side-Side (SS) bending moments ($M_{tn}$ and $M_{tl}$) for each window.

The authors dealt with the methodology to recover the low-frequency fatigue cycles in detail in a previous study (Sadeghi et al., 2022), as such, the process is not repeated in this paper. In that work, three years of SHM data have been used to demonstrate that concatenating the segmented time signals and cycle-counting the resulting signal has the same effect as merging the segmented cycle-count histograms with the cycle-count histogram of the residuals sequence from the segmented data. This operation has multiple advantages: (i) it is faster, (ii) it requires less database space, and (iii) it is extremely more versatile than the short-term signals concatenation (Marsh et al., 2016). The necessary steps to obtain a long-term cycle-



count histogram through the LFFD recovery algorithm are described in detail in our previous publication (Sadeghi et al., 2022). The final result is a cycle-count histogram which contains the low-frequency cycles.

In the current study, the overall fatigue damage with LFFD is then calculated using the py-fatigue python package (D'Antuono et al., 2023) and through the Palmgren-Miner rule (5) combined with the stress-life (S-N) curve as defined by Basquin's law (Basquin, 1910) (4). Although the Palmgren-Miner linear rule is relatively imprecise and does not account for

load sequence or memory effects, it is by far the most popular and commonly utilized damage accumulation rule (Ciavarella et al., 2018), required as a safety standard even by safety-critical industries such as aviation transport (EASA, 2020), although more accurate models are available (Hsiao et al., 2021). Following the DNV-RP-C203 guidelines using the single gradient curves (Iliopoulos et al., 2017), the selected S-N curves are illustrated in Fig.2. We chose three linear S-N curves with different slopes (3, 4, and 5) because these are normally used to calculate damage equivalent moments (DEM), as one

of the applications of the desired LFFD-factor would be to consider the LFFD effect in the calculated DEM (refer to Appendix C). The focus on the single gradient linear S-N curves is motivated by the desire to have a general LFFD-factor, independent of the fatigue spectra and multiplicative factors. On the other hand, bi-linear S-N curves will yield to a non-generalizable LFFD-factor as will be discussed in Sect.4.3, using two bilinear S-N curves of DNV_D_A and DNV_D_CP, which are the S-N curves for detail type D in air, and in seawater with cathodic protection, respectively (Iliopoulos et al.,

2017). Meanwhile, the seawater with free corrosion curve is a single slope S-N curve (m=3), and thus LFFD-factor for m=3 applies.

It is worth mentioning that in the current research, for simplicity no existing damage was assumed on the structures and initially the residuals are counted as half cycles. Furthermore, no mean stress corrections were used and the loading was assumed to be uniaxial. Both assumptions serve to simplify the outcomes of this analysis yet align with DNVGL-RP-C203

(2019). As the recommended practice specifies that if the maximum cyclic stress is greater than zero (overall tension), no mean stress correction shall be applied (factor = 1), while a constant factor < 1 can optionally be used for the stress ranges when residual compressive stresses can be documented.

Similarly, regarding the behavior under combined loading, we again follow the recommended practice (DNVGL-RP-C203, 2019). As the used S-N curves were obtained from specimens that underwent uniaxial tests, therefor we are disregarding the

impact of torsional and shear loads. From a practical point, no data was collected for these loads due to the lack of sensors in those directions.



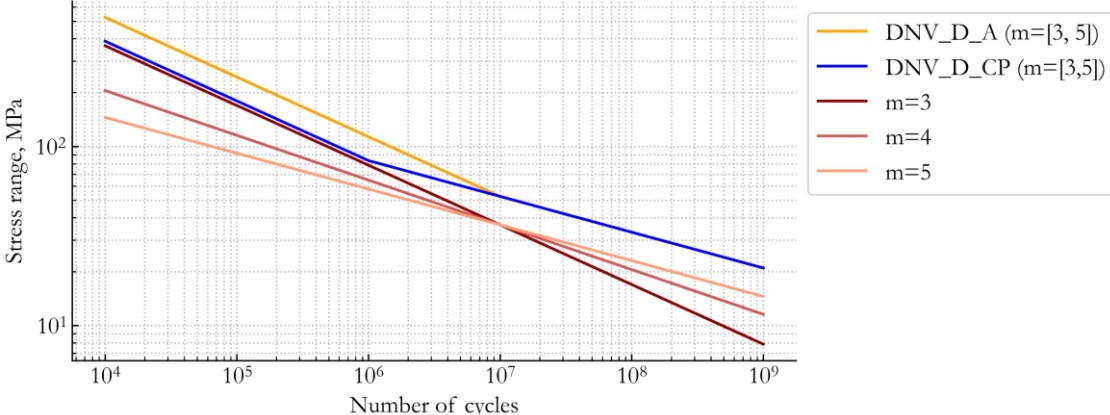

**Figure 2: Used S-N curves.**

The short-to-long-term fatigue damage LFFD-factor is expressed in Eq. (6) as the ratio of the damage from long-term to the
damage from short-term fatigue histograms, i.e.:

$$N \cdot (\Delta\sigma)^m = a, \tag{4}$$

$$D = \sum_{j=1}^{N_{blocks}} \frac{n_j}{N_j} = 1/a \cdot \sum_{j=1}^{N_{blocks}} n_j \, (\Delta\sigma)_j^m, \tag{5}$$

$$LFFD\_factor = D_{LT}/D_{ST}, \tag{6}$$

Where (m, a) are the S-N curve slope and intercept, respectively, $n_j$ is the number of cycles in the j-th load block, $N_j$ are the
cycles to failure at the load level $\Delta\sigma_j$, and $D_{LT}$, $D_{ST}$ are the long-term and short-term fatigue damages. As the number and the
stress range of large cycles increase after the LFFD recovery, see Fig.3, LFFD_factor ≥ 1.

Figure 3 shows how the number of full cycles increases after considering the low-frequency cycles. It can be seen that the
added full cycles have mainly stress ranges of over 10 MPa and therefore, they will have a large contribution to the final
damage.



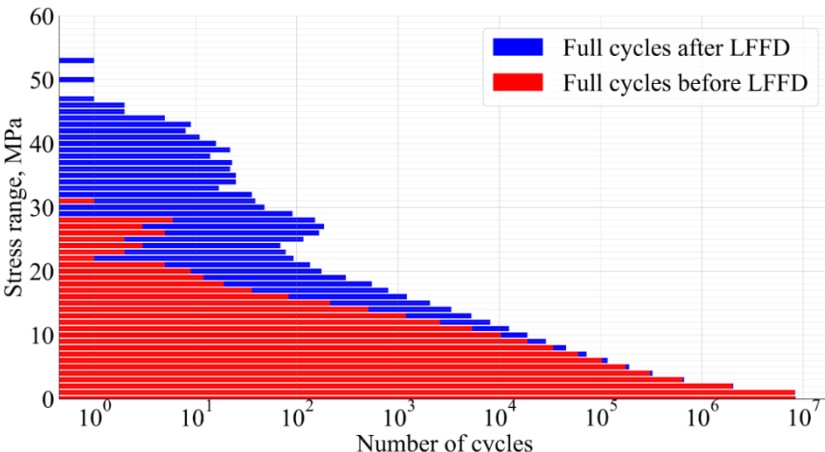


**Figure 3: Comparison of full cycles of one year of cycle counts FA from Turbine1, before and after applying LFFD.**

It is noteworthy that the LFFD-factor for single gradient curves does not depend on the S-N curve intercept (a) but the slope (m) only. This is shown in Eq. (7) by substituting Eq. (5) in Eq. (6):

$$\text{LFFD\_factor} = \frac{\sum_{j=1}^{N_{blocks}} n_{LT,j} \, (\Delta\sigma)_{LT,j}^{m}}{\sum_{j=1}^{N_{blocks}} n_{ST,j} \, (\Delta\sigma)_{ST,j}^{m}}, \tag{7}$$

As a consequence, given that $\max\left[(\Delta\sigma)_{LT,j}\right] \geq \max\left[(\Delta\sigma)_{ST,j}\right]$, if the slope increases, the effect of the large low-frequency cycles will increase, and the LFFD-factor will increase.

In Fig.4 (a), a single day of real-world data collected from Farm1 is used to show how a low-frequency cycle forms. The measurements for both the FA and SS bending moments as well as two individual strain gauges are shown against the wind speed and direction. The vertical grid lines show the 10-minute short-term windows. For the FA direction, slow variations in

the bending moment spanning several hours can be seen. These LFFD far exceed the cycles observed within a 10-minute window and can be directly correlated to the variations in wind speed over the day. Meanwhile, for the SS, the bending moment is not affected by these variations in wind speed.

While the FA bending moment is only affected by the wind speed, a secondary effect can be seen for the individual strain gauges at headings 145° and 265° (S3 and S5). For example at the start of the day, S3 is in tension as the wind is coming

from the South-East (153°), meanwhile, S5 is in compression. In the mid-day, the wind heading is between the two sensors, and therefore, their mean strain is around zero. Towards the end of the day, S3 is in compression and S5 is in tension. These LFFD cycles are readily explained when considering that the wind direction also has changed towards the West (237°). Examples of two LFFD cycles are shown by the black dashed lines. This illustration suggests that the size of the LFFD effect will vary along with the wind conditions at a site as well as the type of signal considered. Please note that in this plot, the

sensors' strain is detrended to have zero means. In Fig.4 (b), the frequency spectrum of the two sensors and the FA and SS strain signals are shown. The sensors' frequency plots are overlying. The first highest peak happens in a very low frequency



of 0.016 Hz and FA and SS have the highest and lowest power density, while sensors power is in between them. The second dominant frequency is around 0.36 Hz at the frequency of the first dynamic mode, but in this frequency, SS has the largest height while FA has the least value.

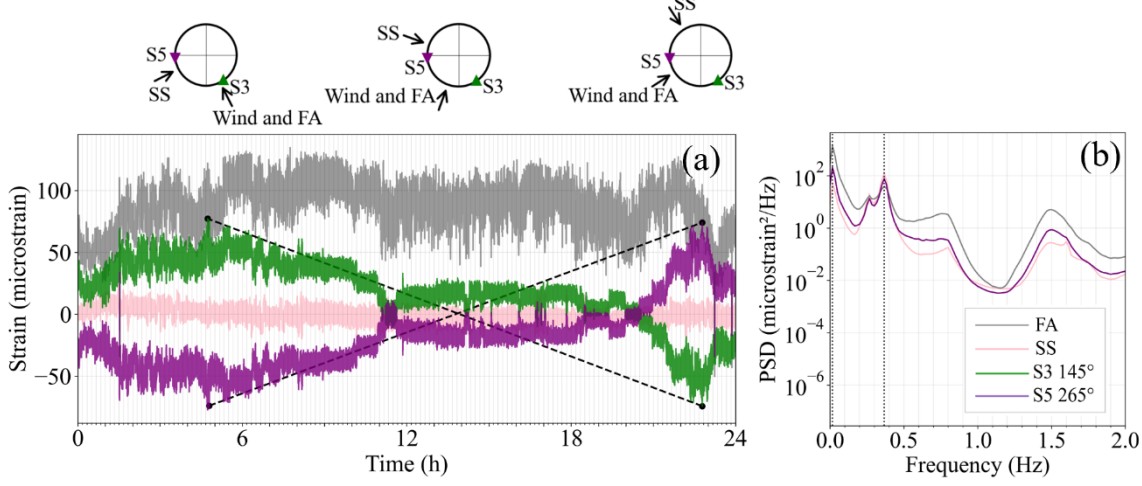


**Figure 4: (a) Strain signals of FA, SS, and two sensors. Black dashed lines show the schematic of two LFFD cycles forming during a day from the change in wind speed and direction. (b) The frequency spectrum of FA, SS, and two sensors. The two vertical dotted lines show the first and second modes.**

## 3 Objective

In the present work, we perform an in-depth analysis of the LFFD-factors to investigate the variability of the LFFD-factor over time, across different sensor positions, locations, and wind farms. We are interested in the additional damage that might be ignored if we do not consider the LFFD effect in the common practices. Our work aims at practical industrial applications and, as such, it must adopt the most widely accepted industry practices and standards for fatigue lifetime prediction. In particular, the following is investigated in further detail:

1.      The trend of the convergence of LFFD-factors over time and the relative importance of growing time horizons in LFFD.

2.      Comparison of LFFD-factors for the direct measurements of sensors along the circumference of the wind turbine support structure with those calculated for FA/SS bending moments.

3.      Considering multiple support structures (with similar designs) in the same wind farm.

4.      Considering multiple foundation designs coming from two different wind farms.

The final scope of this study is to define the right short-to-long-term fatigue damage LFFD-factor for a specific OWT that can be applied as a multiplicative constant to the short-term damage whenever the low-frequency fatigue cycles cannot be retrieved, for instance, when working with 10-minute damages, or equivalently when working with short-term damage





equivalent loads (DELs), where the time order of the short-term dataset is not available (Hübler et al., 2018; Weijtens et al.,
195 2016).

## 4 Case studies

### 4.1 Behavior of LFFD-factor for FA/SS and sensors, and trend over time

Figure 5. a shows the trend of the LFFD-factor over time along the circumference (four sensors) of the mono-pile under
consideration (T1) for single slope S-N curves with slopes of m=3, 4, 5 shown by purple, yellow, and blue lines.
Additionally, the results of Marsh et al. (2016), from one year period of stress data collected from a multi-megawatt offshore
wind turbine support structure are plotted for m=3 and 5.

Generally, the factor depends mostly on the S-N curve slope (m) rather than the considered strain history. The first
stabilization of the LFFD-factor appears after three months, coherently with what was found by Marsh et al. (2016), but after
almost six months, a season-dependent effect causes some variation in the LFFD-factor. For this reason, we consider the
alternation of four seasons (≥ 1 year) as the minimum time window that gives a stable value of LFFD-factor. Concerning the
influence of the S-N curve, and the behavior along the circumference, after convergence, the LFFD-factor is relatively stable.
Its variations can be safely expressed by a mean value ± two standard deviations (SD) (grey area and the grey solid line in
Fig.5 (a)). Marsh et al. (2016) showed a factor of around 2.3 and 2.7 for m=5 and 1.1 for m=3, which are comparable to the
values from the current study. A summary of the information on the converged factors for different S-N slopes (m = 3, 3.5,
4, 4.5, and 5) for all four turbines is available in Appendix A.

To evaluate the different LFFD-factors for FA/SS/strain of individual sensors, we have compared the LFFD-factor calculated
from the FA/SS bending moments with the average LFFD-factor of sensors of Turbine 1, in Fig.5 (b). We only focused on
m=5, as it has the highest LFFD-factor values. The plot shows that the lowest LFFD-factor is found for the SS direction and
that the average LFFD-factor of sensors (2.76) is much higher than those found for the FA and SS directions (1.89 and 1.5,
respectively). As illustrated in Fig.4 the likely cause of the larger role of LFFD for fixed headings is the LFFD cycles caused
by the changing wind directions over time. The maximum margin of scatter for the average value of all four sensors (grey
line in Fig.5 (a)) (for m=5) is 0.25. It is equal to two standard deviations which is shown by the grey area.

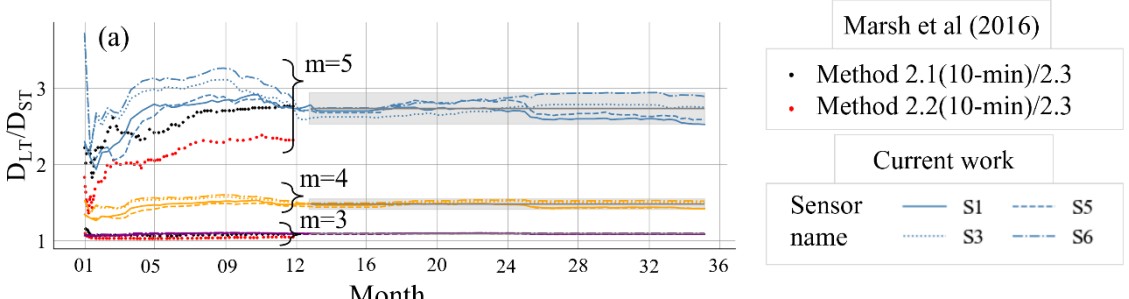





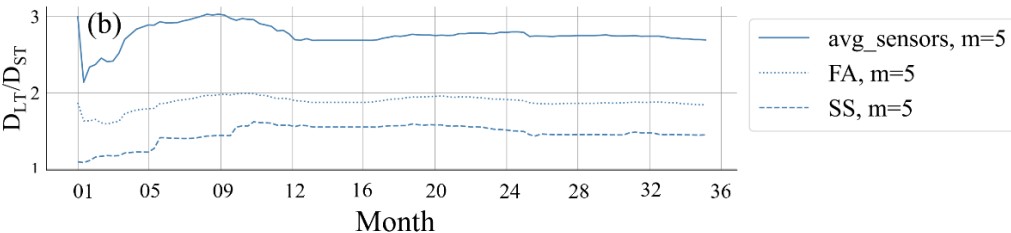

**Figure 5: (a) Trend of LFFD-factor along the circumference of T1 over time using three years of SHM data, plus the results of Marsh et al (2016). The grey solid line shows the average of converged values of all sensors, and the grey area is the average ± 2SD (b) Comparison of LFFD-factor value from FA/SS and the average of sensors for m=5.**

To better understand the mechanism behind the LFFD-factors, we calculated the LFFD-factors by considering different window sizes over the whole measurement duration, yearly, seasonal, monthly, daily, and 10-minute. For example, for the

whole duration, all the low-frequency cycles are accounted, for yearly (Y), cycles shorter than one year are considered, for seasonal (Q), cycles shorter than three months, for monthly (M), cycles below one month, and for daily (D), cycles shorter than a day, and (T) without accounting for LFFD in 10-minute windows. Figure 6 shows the results for damages calculated with the mentioned aggregation windows. The damages are normalized to the "total damage" from cycle-counting the entire signal together. The result was valid over both farms and all four turbines, therefore we only show the T1 results. The plots

for the other three turbines are in Appendix B.

Note that the LFFD-factor(s) are found as 100/Damage("T"). Figure 6 illustrates that for all slopes, daily cycles contribute to the majority of the LFFD-factor. For this turbine, we notice that for m=3, cycle-counting on a daily basis covers ca. 98 % of the "total damage" of an individual sensor and in the FA direction. By comparison, for m=4, cycle-counting on a daily basis only accounts for ca. 90 % of the "total damage" for individual sensors, while 96 % and 99 % of total damage are accounted

for when respectively a weekly and a monthly basis is considered. For m=5, cycle-counting on a monthly basis covers ca. 96 % of the "total damage" (of sensors). Results seem to suggest that considering longer periods has trivial added value. The exception is an apparent large contribution of yearly cycles in the SS result for m=5, however, this behavior seems limited to turbine T1 and might be due to an anomalous cycle, e.g. due to a sensor malfunction.

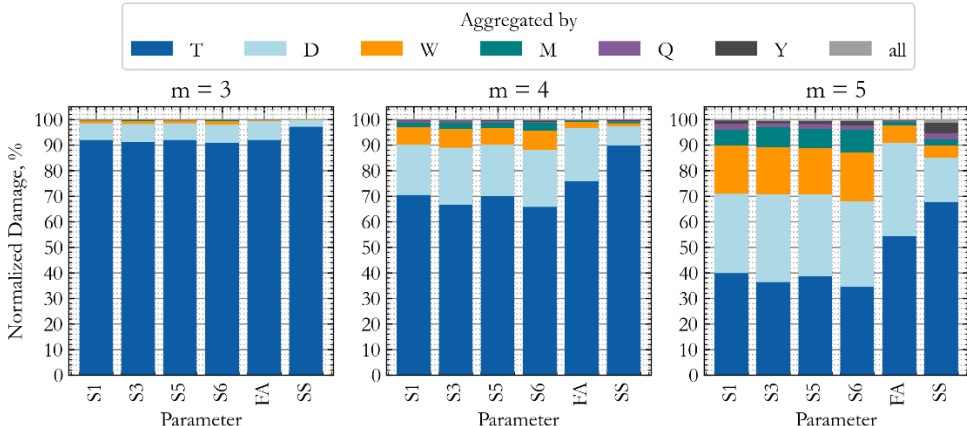

Figure 6: Share of low-frequency cycles of various lengths in the final LFFD-factor of T1 after three years. Damages are normalized to 100 %. (Without LFFD: "T", Daily: "D", Weekly: "W", Monthly: "M", Quarterly: "Q", and Yearly: "Y"). Results for the three other turbines are provided in Appendix B.

Consistent with Eq. (7), by the increase in the slope of the S-N curve, the share of long-term damages and the need to consider a longer time window increases. Among the sensors, S3 and S6 which are normal to the dominant wind direction have the smallest short-term damage contribution, which translates to the highest LFFD-factors.

Comparing the composition of the LFFD for individual sensors against that of the projected FA and SS directions, we observe that the share of cycles lasting more than a day is nearly absent in both FA and SS. Meanwhile, these long cycles do play a role when considering the strain history of a sensor for a fixed location on the structure. Considering that the wind direction changes much slower than the wind speed (it usually takes more than a day to have a change in the wind direction), the shares of weekly and monthly cycles are higher for individual sensors due to the additional cycles from wind direction changes which by definition have been taken out of the projected FA and SS signals. Meanwhile, it is observed that, for all values of m, the impact of the LFFD cycles is the smallest for the SS direction and can be even considered negligible for m=3. This can be explained as the main loading in this direction comes from cyclic loads of (misaligned) waves and the tower dynamics with periods of less than 10s, rather than from the much slower variations in wind speed or wind direction. Also, note that the variability among the LFFD-factors of sensors is less than the difference between sensors and the FA and SS LFFD-factors.

## 4.2 Behavior of LFFD-factor on different wind turbines within the same farm and among two distinct farms

Figure 7 shows the average of LFFD-factors for each slope of the S-N curve (m=3, 4, and 5) after convergence (> 9 months). In each box, the factors for all four turbines across the two farms are provided.

If we take the average of sensors for each turbine, the difference between turbines in the same farm is below 0.02, 0.1, and 0.3 for m=3, 4, and 5, respectively (refer to Appendix A). The wind and wave conditions for the three turbines in the same farm are quite similar. Therefore, the difference from one turbine to another might be explained by the different reactions of



the turbine's structure to these loading conditions. T2 has the lowest LFFD-factors because this turbine experienced more parked conditions compared to T3 and T4.

In parked conditions the importance of wave-induced loading increases, while the low-frequency wind loads are reduced significantly, ultimately reducing the LFFD damage. Moreover, the fatigue loading is larger in parked conditions, as a result, T2 has accumulated more base damage than T3 and T4. The higher base damage (without LFFD) and lower added LFFD damage will lead to a lower LFFD-factor.

FA LFFD-factor is slightly higher than those for sensors for m=3, but with the increase in the m, sensors' LFFD-factors
overcome FA. That is because, with the increase of the S-N slope, the added LFFD damage due to the wind direction grows. SS has always the lowest LFFD-factors. The LFFD-factor of FA and SS can be up to around (1.1, 1.4, and 2) and (1.03, 1.1, and 1.5) for m=3, 4, and 5, respectively.

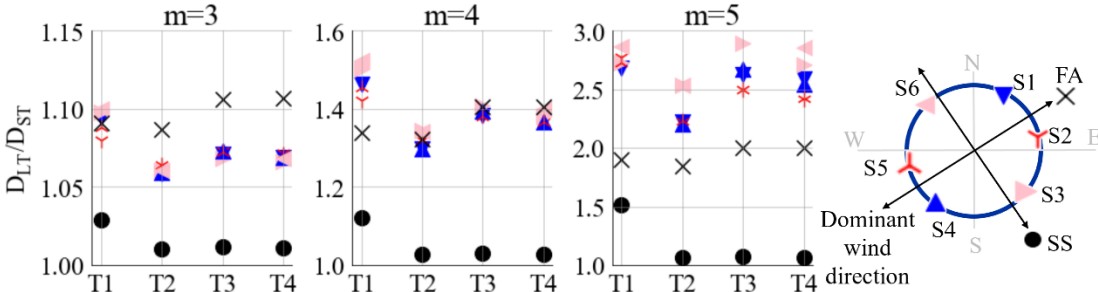

**Figure 7: Average of the stabilized LFFD-factors after nine months for the six positions along the circumference and FA and SS,**
**for T1 of Farm1, and T2, T3, and T4 of Farm2. Each box is for an S-N curve slope.**

Since Farms 1 and 2 significantly differ in design and natural frequencies, a comparative study helps to understand if and how these differences can affect the short-to-long-term fatigue damage LFFD-factor behavior and value. From Fig.7, it seems overall that the low-frequency fatigue effect is different between the two farms, with higher LFFD-factors for Farm1 for all values of m. If we take the average LFFD-factor of sensors for each turbine, the difference between turbines in the
two farms is below 0.05, 0.2, and 0.5 for m=3, 4, and 5, respectively (refer to Table 1 in Appendix A). Conversely, T2, T3, and T4 are larger structures (resonance frequency is lower and therefore closer to the wave frequency), the damage is more affected by waves, and wave loading gives less low-frequent variation in damage.

For each S-N slope, the variability of the LFFD-factor is less between sensors compared with FA and SS. For sensors and SS, the shift of the turbine in a farm causes less change in the factor compared to the shift in the farm. While for FA, the shift
in the farm yields a greater change of factor, compared to the change of turbine in a farm.

As expected, opposite sensors show similar, if not precisely identical, LFFD-factors due to the circular symmetry of the support structure. In all four turbines, the sensors perpendicular to the dominant wind (S3 and S6) have higher LFFD-factors. Surprisingly, the sensors that are mostly affected by the thrust loading (S1-S4, and S2-S5 pairs) have lower LFFD-factors. We observed that among the sensors normal and parallel to the dominant wind direction, the differences in base damage



("T") are larger than the differences in the added LFFD damage. Therefore, the LFFD-factor of sensors parallel to the wind is lower than the sensors normal to the wind.

**4.3 Role of LFFD with bi-linear S-N curves**

In this study, until here we analyzed the effect of linear S-N curves with slopes equal to m=3, 4, and 5 on the LFFD-factor. These S-N curves are mainly used to calculate the damage equivalent moments (DEM). But to calculate the fatigue damage
of OWTs, bi-linear S-N curves are widely used. In Fig.2, two commonly used bi-linear curves from DNVGL-RP-C203 (2019) are selected to study the change in the LFFD-factor with the use of bi-linear instead of linear S-N curves. Both S-N curves are partially at slopes m=3 and m=5, respectively for large and small stress cycles. As a result, the bi-linear S-N curve will penalize large cycles, such as those caused by LFFD, less than a single sloped m=5 curve. A fundamental consideration is that for bi-linear curves the LFFD factor will be dependent on the stress ranges. Depending on whether the majority of
stress cycles fall in the m=3 or m=5 region, it will result in respectively a lower or higher LFFD factor. In the study shown in Fig.8, the LFFD-factor of four turbines was analyzed after one year with the aforementioned bi-linear S-N curves. To investigate the role of the overall stress range, a stress concentration factor (SCF) is introduced in Eq. (8):

$$\Delta\sigma_{SCF} = \text{SCF}.\Delta\sigma, \tag{8}$$

In Fig.8, the SCF is varied from 1 to a maximum of 4, and LFFD-factors are calculated for each value of SCF. Note that the
results hold for any (combined) multiplier on the stress ranges, e.g. a material factor (Natarajan, 2021). Figure 8 shows that as the SCF increases, the LFFD-factor decreases. This is because the higher SCF shifts the fatigue spectra upwards towards a higher stress range, into the m=3 region of the bi-linear S-N curve. As a result, pushing the LFFD-factor closer to the LFFD-factor of the linear S-N curve with m=3. However, even with an SCF of 4, the LFFD-factors of sensors and FA from the bi-linear curves were not as low as those from the linear S-N curve with m=3. On the other hand, the bi-linear S-N curve
generally resulted in a lower LFFD-factor compared to the linear S-N curve with m=5, even when the SCF was 1. Due to the lower stress ranges, which are mostly affected by the m=5 region and do not reach the m=3 part of the bi-linear curve at all, there is a significantly less difference between the linear and bi-linear LFFD-factors for the smallest turbine (T1) compared to other turbines. Thus, the fatigue spectrum continues to be in the m=5 region of the bi-linear curve even after applying an SCF of 4.

Furthermore, the LFFD-factors from the S-N curve with cathodic protection (DNV-D-CP) are higher than those from the S-N curve in the air (DNV-D-A). This is because the transition of the DNV-D-CP curve to m=3 is at higher stress ranges, making the LFFD-factors closer to the LFFD-factors from the linear S-N curve with m=5. One interesting note is the FA LFFD-factor of T1 for DNV_D_CP, which from SCF=1 to 2, does not have the same sudden drop compared with other cases. This is explained as the full fatigue spectrum remains in the m=5 region even with SCF=2.





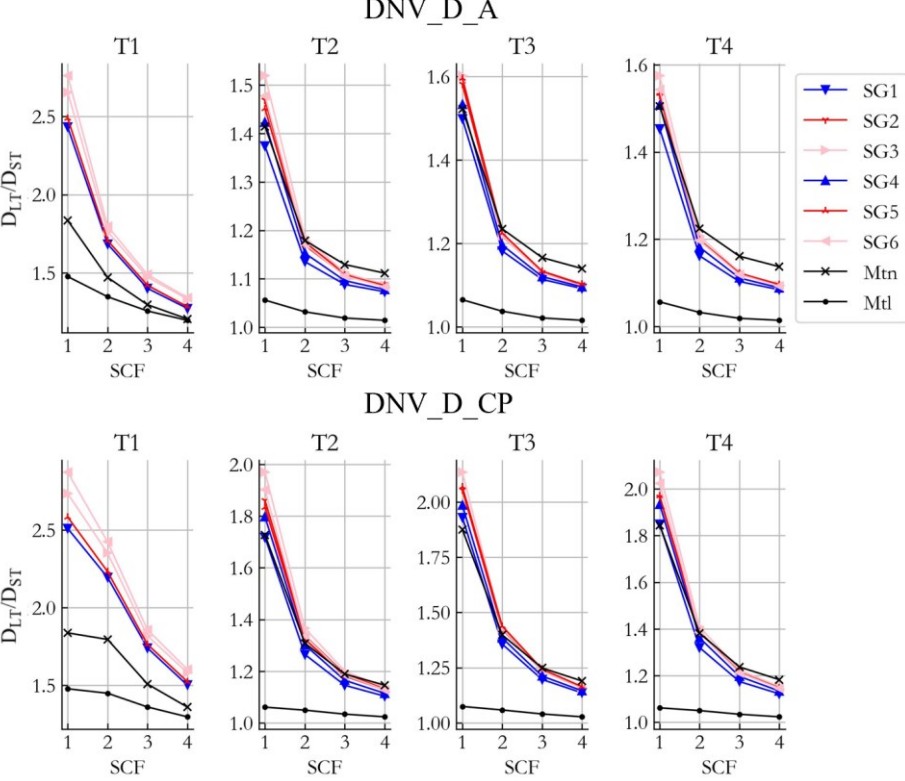

**Figure 8: Trend of LFFD-factor versus SCF for all turbines for both DNV-D-A and DNV-D-CP.**

**5 Conclusions**

This study examined the long-term OWTs fatigue using three years of SHM data. We found that at least one year is needed to achieve a reliable LFFD-factor. We observed that the LFFD-factor is lowest for the SS direction and mainly highest for fixed headings. An effect that strengthens for larger S-N slopes. Considering the effect of gradual variabilities (LFFD) in the analysis can contribute significantly to damage, up to 65 % for an individual sensor and a Basquin slope of m=5.

The slower variation of wind direction compared to wind speed leads to larger importance of weekly and monthly cycles in overall fatigue damage for individual headings compared to the FA/SS directions. The majority of the low-frequency cycles last less than a day and the share of low-frequency cycles in the total damage increases for higher slopes in the S-N curve.

Fatigue analyses on the strain sensors showed that the heading has a secondary effect on the LFFD-factor.

The LFFD-factor calculated from one turbine can be roughly used for other turbines within the same farm. However, LFFD-factors for different wind farms with different support structure designs can vary up to 50 % and 110 % for linear S-N curves with m=5 and bilinear S-N curves with m=[3,5], respectively. The bilinear curves cannot produce conclusions that are generalizable since the LFFD-factor varies from spectrum to spectrum.



In future work, we plan to use the LFFD-factors whenever the recovery of the low-frequency fatigue from cycle count histograms cannot be directly applied, e.g., when only DELs are available (cf. Appendix C) or when we lose the time sequence in the data (like bootstrapping or binning the short-term damages).

**Team list**

Negin Sadeghi, Pietro D'Antuono, Nymfa Noppe, Koen Robbelein, Wout Weijtjens, Christof Devriendt.

**Author contributions**

NS: Analyzing, Interpretation, Writing. PD: Programming Support,  Expert Opinion. NN: Supervision, Conceptualization. KR: Interpretation. WW: Validation, Data Preparation, Interpretation. CD: Supervision, Data Collection, Validation. All authors contributed to Review and Editing.

**Competing interests**

The authors declare that they have no conflict of interest.

**Data availability**

Seeing that the data are proprietary to the industrial partner of this project, the data used in this paper cannot be made publicly available.

**Acknowledgment**

This research is conducted within the project MAXWind, funded by the Belgian Energy Transition Fund (ETF).

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

## Appendix A

Table A1 lists the observations of the converged LFFD-factors after nine months from Fig.5. a and for the three turbines of Farm2. In addition, the values for m = 3.5 and 4.5 are included. The highest LFFD-factors for m=3, 4, and 5 are, as you can see, approximately 1.1, 1.5, and 2.9, respectively. And for m=3, 4, and 5, the maximum 2SDs are roughly 0.02, 0.1, and 0.5, respectively.

**Table A1 The converged LFFD factors for sensors, FA and SS, for all four turbines.**

|  |  | T1 | | T2 | | T3 | | T4 | |
|---|---|---|---|---|---|---|---|---|---|
|  |  | average | 2SD | average | 2SD | average | 2SD | average | 2SD |
| m=3 | S1 | 1,094 | 0,010 | 1,059 | 0,003 | 1,071 | 0,003 | 1,069 | 0,004 |
|  | S2 |  |  | 1,064 | 0,004 | 1,075 | 0,007 | 1,072 | 0,002 |
|  | S3 | 1,098 | 0,005 | 1,061 | 0,004 | 1,068 | 0,003 | 1,066 | 0,002 |
|  | S4 |  | 0,000 | 1,059 | 0,003 | 1,073 | 0,003 | 1,069 | 0,003 |
|  | S5 | 1,088 | 0,008 | 1,064 | 0,004 | 1,073 | 0,003 | 1,071 | 0,002 |
|  | S6 | 1,100 | 0,005 | 1,061 | 0,004 |  | 0,000 | 1,068 | 0,003 |
|  | SENSORS | 1,094 | 0,012 | 1,061 | 0,005 | 1,072 | 0,006 | 1,069 | 0,005 |
|  | FA | 1,091 | 0,009 | 1,087 | 0,006 | 1,106 | 0,003 | 1,107 | 0,003 |




| | | | | | | | | |
|---|---|---|---|---|---|---|---|---|
| | SS | 1,029 | 0,002 | 1,011 | 0,000 | 1,012 | 0,000 | 1,011 | 0,000 |
| m=3.5 | S1 | 1,219 | 0,025 | 1,137 | 0,009 | 1,174 | 0,011 | 1,169 | 0,013 |
| | S2 | | | 1,147 | 0,009 | 1,173 | 0,008 | 1,169 | 0,007 |
| | S3 | 1,235 | 0,014 | 1,147 | 0,009 | 1,168 | 0,008 | 1,161 | 0,007 |
| | S4 | | | 1,137 | 0,008 | 1,178 | 0,010 | 1,167 | 0,012 |
| | S5 | 1,209 | 0,021 | 1,147 | 0,009 | 1,173 | 0,008 | 1,168 | 0,008 |
| | S6 | 1,242 | 0,015 | 1,146 | 0,009 | | | 1,167 | 0,007 |
| | SENSORS | 1,226 | 0,018 | 1,144 | 0,009 | 1,173 | 0,009 | 1,167 | 0,009 |
| | FA | 1,186 | 0,019 | 1,175 | 0,010 | 1,219 | 0,006 | 1,223 | 0,009 |
| | SS | 1,059 | 0,007 | 1,017 | 0,001 | 1,019 | 0,001 | 1,017 | 0,001 |
| m=4 | S1 | 1,465 | 0,055 | 1,299 | 0,016 | 1,385 | 0,016 | 1,372 | 0,022 |
| | S2 | | | 1,321 | 0,025 | 1,394 | 0,041 | 1,369 | 0,016 |
| | S3 | 1,504 | 0,033 | 1,344 | 0,027 | 1,406 | 0,012 | 1,381 | 0,011 |
| | S4 | | 0,000 | 1,298 | 0,017 | 1,396 | 0,015 | 1,367 | 0,020 |
| | S5 | 1,454 | 0,048 | 1,321 | 0,026 | 1,383 | 0,021 | 1,368 | 0,017 |
| | S6 | 1,523 | 0,037 | 1,343 | 0,027 | | 0,000 | 1,400 | 0,012 |
| | SENSORS | 1,480 | 0,077 | 1,321 | 0,045 | 1,393 | 0,031 | 1,376 | 0,029 |
| | FA | 1,340 | 0,034 | 1,324 | 0,025 | 1,404 | 0,012 | 1,405 | 0,010 |
| | SS | 1,121 | 0,020 | 1,027 | 0,002 | 1,030 | 0,001 | 1,027 | 0,001 |
| m=4.5 | S1 | 1,917 | 0,104 | 1,625 | 0,033 | 1,831 | 0,041 | 1,814 | 0,053 |
| | S2 | | | 1,655 | 0,062 | 1,798 | 0,056 | 1,764 | 0,050 |
| | S3 | 1,977 | 0,082 | 1,753 | 0,064 | 1,897 | 0,050 | 1,828 | 0,037 |
| | S4 | | | 1,616 | 0,030 | 1,853 | 0,037 | 1,798 | 0,049 |
| | S5 | 1,919 | 0,095 | 1,658 | 0,067 | 1,795 | 0,057 | 1,764 | 0,053 |
| | S6 | 2,034 | 0,094 | 1,748 | 0,062 | | | 1,886 | 0,038 |
| | SENSORS | 1,962 | 0,094 | 1,676 | 0,053 | 1,835 | 0,048 | 1,809 | 0,047 |
| | FA | 1,574 | 0,055 | 1,539 | 0,029 | 1,666 | 0,013 | 1,672 | 0,023 |
| | SS | 1,254 | 0,049 | 1,042 | 0,005 | 1,048 | 0,004 | 1,043 | 0,003 |
| m=5 | S1 | 2,681 | 0,184 | 2,225 | 0,051 | 2,629 | 0,087 | 2,591 | 0,112 |
| | S2 | | | 2,235 | 0,130 | 2,518 | 0,122 | 2,426 | 0,108 |
| | S3 | 2,727 | 0,180 | 2,541 | 0,151 | 2,897 | 0,057 | 2,713 | 0,060 |
| | S4 | | 0,000 | 2,207 | 0,051 | 2,676 | 0,087 | 2,553 | 0,106 |
| | S5 | 2,724 | 0,178 | 2,240 | 0,143 | 2,502 | 0,123 | 2,429 | 0,116 |
| | S6 | 2,868 | 0,211 | 2,540 | 0,145 | | 0,000 | 2,861 | 0,074 |
| | SENSORS | 2,760 | 0,256 | 2,322 | 0,349 | 2,637 | 0,323 | 2,586 | 0,360 |
| | FA | 1,897 | 0,080 | 1,844 | 0,048 | 2,004 | 0,019 | 2,002 | 0,022 |
| | SS | 1,509 | 0,129 | 1,063 | 0,006 | 1,074 | 0,004 | 1,064 | 0,005 |

**Appendix B**

Figure 9 shows the share of low-frequency cycles with multiple aggregation window lengths. As shown, the shares of LFFD damages among the same farm are similar, except for the lower share of T2, since it was more in parked condition. In general, the share of LFFD damage (except for SS) is almost similar for T1 and other turbines.



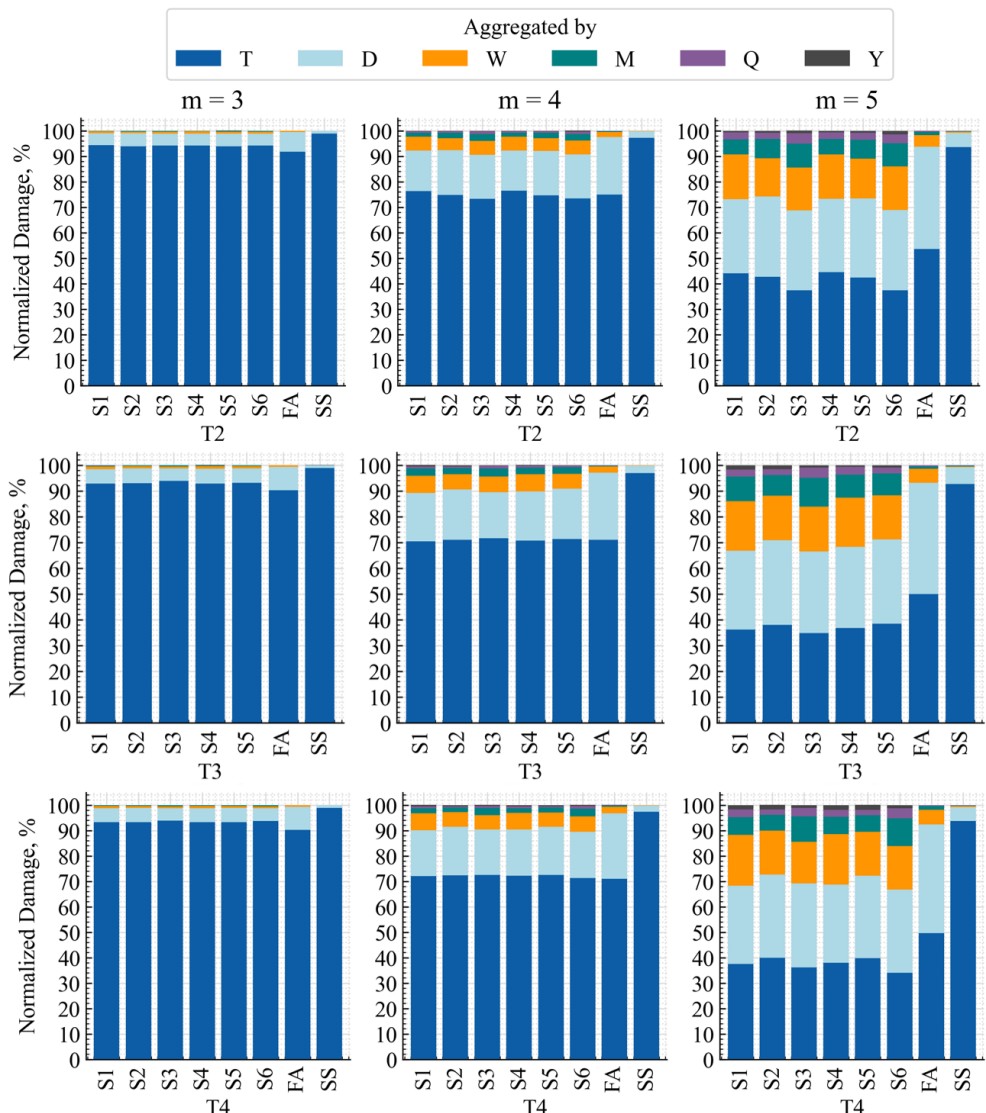

**Figure B1: Share of low-frequency cycles with different cycle periods for turbines of Farm2.**

## Appendix C

The damage equivalent load (DEL) derived from a load time signal history for a chosen single slope S-N curve (typically m=3, 4, 5 for steel) and an equivalent number of cycles (typically $N_{eq}$=1e7) creates the same amount of fatigue damage as the Miner damage derived from original load time signal on the same S-N curve. The DEL is a direct quantification of fatigue loads derived from load measurement time series. The DEL from simulations is made during the design phase under the condition that the S-N curve characteristics (m and $N_{eq}$) are the same. The DEL can be utilized as an alternative for damage because the Miner damage cannot be measured physically and requires a more refined selection of the S-N curve.





We use DELs with S-N slopes of m=3, 4, and 5 in accordance with established practices3. Considering the logarithmic distribution of damage, DEL is often employed in the design process of (offshore) wind turbines since it offers a linear scale for presenting damage. This choice assures independence from the S-N curve's intercept and improves readability.

In this appendix, we show how the short-to-long-term fatigue damage LFFD-factor, could be applied to convert a short-term DEL into a long-term DEL. For the demonstration, we need the definitions of DEL (Natarajan, 2021) (or, more specifically, DES, as it is a damage-equivalent stress range), the S-N curve in Basquin's law, and the Palmgren-Miner rule.

$$\text{DES} = \left(\frac{\sum_{j=1}^{N_{blocks}} n_j (\Delta\sigma)_j^m}{N_{eq}}\right)^{1/m}, \tag{C8}$$

$$(\Delta\sigma)^m = a/N, \tag{C9}$$

By substituting Eq. (C9) into Eq. (C8), one retrieves the definition of DES concerning the Palmgren-Miner rule.

$$\text{DES}^m \cdot N_{eq} = \sum_{j=1}^{N_{blocks}} n_j (\Delta\sigma)_j^m = a \cdot \underbrace{\sum_{j=1}^{N_{blocks}} n_j/N_j}_{\text{PM rule}}, \tag{C10}$$

Therefore:

$$\text{DES}^m \cdot N_{eq} = a \cdot D, \tag{C11}$$

Applying the definition of LFFD-factor leads to:

$$D_{LT} = \text{DES}_{LT}^m \cdot \frac{N_{eq}}{a} = D_{ST} \cdot \text{LFFD\_factor} = \left(\text{DES}_{ST}^m \cdot \frac{N_{eq}}{a}\right) \cdot \text{LFFD\_f}, \tag{C12}$$

From which the relation between $\text{DES}_{LT}$ and $\text{DES}_{ST}$ is straightforward:

$$\text{DES}_{LT} = \text{DES}_{ST} \cdot \text{LFFD\_factor}^{1/m}, \tag{C13}$$