# Peer review of "Quantifying the effect of low-frequency fatigue dynamics on offshore wind turbine foundations: a comparative study"

_Wind Energy Science, 2023_

## Author Comment (AC1)

Dear Reviewer,
Thank you for taking the time and effort to review our article. We appreciate your valuable feedback and suggestions.

The authors present a very relevant and interesting analysis into quantifying the impact of low frequency cycles on fatigue damage.

While the subject is clearly presented, the results are not convincing. The following aspects are to be clarified or added before the article can be published.

1) The low cycle region of the SN curve for welded steel is conventionally taken with the slope m = 3 as given in Eurocode 3, DNV GL C 203 and IIW standards. The knee point is usually 2e06 or higher cycles. Therefore the impact of LFFD would be limited to m =3 or low slope portion of the bi-linear SN curve only and have a smaller contribution relative to the high slope segment of the SN curve. Reference Larsen, G C. Thomsen, K that you quote also states the impact with the small slope only has a 1% increase due to LFFD. In this light Figure 3 needs to be better explained as to how much impact to the Miner sum is obtained when including LFFD.

We understand the reviewer's perspective, and while there is definitely truth to their statement, our paper aims to provide a more generalized overview of the impact of LFFD. Meanwhile, we believe a similar conclusion can be derived from our current work. With just the minor consideration that most designs will keep loads low and therefore have the majority of the load cycles in the m=5 region, especially those associated with the first tower mode and waves.

As shown in Fig. 2. we covered very common S-N curves including the single-sloped S-N curves with slopes of 3, 4, and 5 which are usually used in the design to calculate DELs, and bi-linear S-N curves with the slopes of 3 and 5 which are the common curves for fatigue damage calculation. Note that even though m=4 is not linked to any design code, it is a commonly used compromise in design for DEL calculation. In Fig. 3, the full cycles before and after considering the low-frequency fatigue Dynamics (LFFD) are shown to illustrate the full cycles that are neglected in case of no LFFD consideration.

To show how much impact on the Miner sum is obtained when including LFFD, Fig. 5 is showing the increase in the damage before and after LFFD, by showing the ratio. There, we can see that after one year, for example, for m=3, 4, or 5, how much percent increase we have on the damage, due to LFFD. In line with the reviewer's comment, the impact of LFFD for m=3 is almost negligible.

As explained in the paper, "It is noteworthy that the LFFD-factor for single gradient curves does not depend on the S-N curve intercept (a) but the slope (m) only."

Also mentioned in the paper: "The focus on the single gradient linear S-N curves is motivated by the desire to have a general LFFD-factor, independent of the fatigue spectra and multiplicative factors. On the other hand, bi-linear S-N curves will yield to a nongeneralizable LFFD-factor".

Therefore, concerning bi-linear curves, we showed in Figure 8 that the LFFD-factor is not fixed, as it is really site dependent, and based on the position of the stress ranges with respect to the knee point, the position of the knee point, and the Stress Concentration Factors (SCF), the LFFD-factor will change. Therefore, based on the place of the knee point, the contribution of the m=3 region and m=5 region can be different, but we cannot say that the higher slope region has no effect from the LFFD. While as the respected reviewer mentions, there might be sites with insignificant/significant LFFD-factor in the condition of using bi-linear curves. Indeed, when a higher SCF is used, the bigger fraction of the load spectrum moves into the m=3 region and the LFFD effect becomes more similar to m=3 (and therefore less). However, as most designs keep loads relatively low, the majority of loads will be in the m=5 region of the S-N curve (hence the common use of m=5 DELs in design). Note this might be counterintuitive, as with a larger SCF, the relative effect of LFFD becomes less.

Regarding the work of Larsen, G C. Thomsen, K, they studied a turbine of 150KW, and the insignificant contribution of LFFD is potentially due to the small size of the turbine. We think it is fair to assume that

conclusions of that generation need not necessarily translate to the current Multi-MW offshore wind turbines. The onshore example of a 2MW turbine from Pacheco et al. (2022) and an offshore example of a multi-MW turbine by Marsh et al. (2016), are more in line with our results, and they all showed that we might have a significant LFFD effect, but there can still be cases that the effect is negligible.

2) Figure 4: Can you show that the 0 load response of the strain sensors have no frequency components (that is noise)?

We suppose that the reviewer is concerned about the fact that cyclic noise in the strain measurements might change the result of LFFD. To show that the sensors have no frequency components under zero load conditions, we do not have a period of measurement with zero loading as the turbine is constantly under wind and wave load. So we cannot provide the reviewer with a plot that shows the Power Spectral Density (PSD) of sensors for a non-loaded period. But by looking at Fig. 4, we observe that the spectrum is dominated by the physical loads, and therefore, we do not doubt that the sensors might have unphysical frequency components.

3) If the stress cycle has a period of several hours or days, then the mean wind speed would have significant changes during that period and the conventional method of fatigue damage accumulation cannot be applied. How is fatigue damage accumulation to be made over different mean wind speeds considering LFFD? Does it require a non-stationary statistics process to compute this?

Although the low-frequency cycles are the results of gradual change in primarily the wind speed and wind direction, our methodology is data-based and the measured data is actually the result of all those loading variations over time. Thus, while the LFFD is a direct result of variation in the wind speed, we did not calculate LFFD based on an assumed wind speed distribution or any other probabilistic method that considers the wind speed variation in time. The only assumption we made for environmental conditions in the processing was that the yaw angle is fixed during 10min windows to calculate the fore-aft and side-side bending moments. This assumption was not made when calculating the damage based on one single strain sensor.

We acknowledge that because we made no assumptions about the wind speed and we merely used the measurement data, our results are site-specific. Therefore, based on the method used in our paper, if the measurement is from a hypothetical site with almost constant wind speed and direction, the LFFD effect would be negligible. However, the investigated sites are inside the most populated area for offshore wind (North Sea), so the results are representative of a sizable offshore capacity.

One should employ different techniques beyond the scope of this article if they want to connect LFFD to wind speed statistics. As far as we are aware, even selecting a Weibull distribution as a typical wind speed distribution does not give information about how wind speed and direction change over time. As a result, additional characteristics are required to demonstrate the randomness of the variation in wind speed over time. E.g. consider following the thought experiment, draw wind speed samples from a Weibull, but then sort them. While the data obtained still adheres to the Weibull distribution, only a very limited LFFD effect is present as just a single cycle in windspeed is experienced.

4) Usually for offshore structures, it is the welded joints that have the lowest fatigue life. It appears these are not considered at all in the present work and therefore is is unclear if LFFD has any impact on design life. The stress at the welded joint is significantly increased due to local stress gradients in different directions. The impact of the stress gradients can be much higher than the increase in loading due to LFFD.

We agree with the reviewer that the critical points have higher damage. We usually calculate their damage from measured data by applying different SCF factors to the stress ranges from the sensors. Although we did not mention any specific weld on the substructure, the paper still covers all points. The majority of the paper is discussing the LFFD-factor with single-slope S-N curves. As mentioned in the article: "We chose three linear S-N curves with different slopes (3, 4, and 5) because these are normally used to calculate damage equivalent moments (DEM), as one of the applications of the desired LFFD-factor would be to consider the LFFD effect in the calculated DEM (refer to Appendix C). The focus on the single gradient linear S-N curves is motivated by the desire to have a general LFFD-factor, independent of the fatigue spectra and multiplicative factors.

As mentioned before, the share of LFFD on the damage on any point of the substructure would be the same, in the case of single-slope S-N curves. As shown in Eq. 1, the applied SCF is canceled out and the LFFD-factor would be fixed for different SCF and therefore different welds.

$$\text{LFFD\_factor} = D_{LT}/D_{ST} = \frac{1/a \cdot \sum_{j=1}^{N_{blocks}} n_{LT,j} (SCF \times \Delta\sigma)_{LT,j}^m}{1/a \cdot \sum_{j=1}^{N_{blocks}} n_{ST,j} (SCF \times \Delta\sigma)_{ST,j}^m} = \frac{\sum_{j=1}^{N_{blocks}} n_{LT,j} (\Delta\sigma)_{LT,j}^m}{\sum_{j=1}^{N_{blocks}} n_{ST,j} (\Delta\sigma)_{ST,j}^m} \qquad Eq.1$$

Where (m, a) are the S-N curve slope and intercept, respectively, $n_j$ is the number of cycles in the j-th load block, $N_j$ are the cycles to failure at the load level $\Delta\sigma_j$, and $D_{LT}$, $D_{ST}$ are the long-term and short-term fatigue damages.

On the other hand, bi-linear S-N curves will yield to a nongeneralizable LFFD-factor". Since often bi-linear S-N curves are applicable for welded joints, we included these in the paper too. As already mentioned above, the LFFD-factor cannot be generalized in this case and will depend on the SCF applied and the knee point of the required S-N curve.

Can an analysis be shown as to how much reduction in lifetime is present at a welded joint due to LFFD ?

As an example, if we consider the yearly damage in the critical welded joint as D, which is calculated by using a single slope S-N curve with m=3, a linear extrapolation of the lifetime without LFFD is $\frac{1}{D}$ years, while the lifetime considering LFFD would be $\frac{1}{D \times LFFD\_factor}$ years. If we read the LFFD_factor for m=3 as 1.09, it means that the lifetime with LFFD is $\frac{1}{1.09} = 91\%$ of lifetime without LFFD, so almost 9% reduction in the lifetime is due to considering the LFFD effect in any weld on the substructure.

In case of bi-linear curve, as the reviewer correctly mentioned, it is not generalizable, since it depends on the position of the stress ranges with respect to the knee point, the position of the knee point, the exact geometry of the detail, and the SCF (as discussed in Section 4.3 of the paper). Therefore, although we showed that with the increase in the SCF, the LFFD effect decreases (for bi-linear S-N curves), yet, we did not claim that the results are generalizable for any specific weld.

5) In figure 5, is the m = 5 slope also at cycles less than 2e06, that is, the minimum slope m = 5?

Yes, in Fig. 5, we assumed linear S-N curves. We did not consider the Haibach rule, as it will introduce a bi-linear curve.

If this is the case, the using Haibach rule, the higher slope of the SN curve would be 9 and would result in higher partial safety factors. Can you quantity what is the impact of the LFFD in the usage of partial safety factors (PSF) in the fatigue life assessment? Does the inclusion of uncertainty due to LFFD result in a significant increase in the PSFs? This assessment is needed to understand its impact in the design process.

We appreciate the question, while we need to ask the reviewer to clarify it to us. At this stage, we do not know what you would like us to undertake. Without more information from the design background, we are unable to establish any connection between the LFFD-factor and the Partial Safety Factors (PSF), since we do not know the precise justifications for the partial safety factors, how they were chosen throughout the design process, or what uncertainties they are intended to cover.

In this work, we quantified the effect of LFFD from a deterministic point of view for a data-driven method. We do not know how the LFFD would change the PSF in the design and as far as we know LFFD is not considered in design by default. It is worth highlighting that the LFFD-factor is applied on the damage while the PSF are applied on the stress ranges. So, translating LFFD-factor to a PSF should be done cautiously, as LFFD-factor is dependent on the used S-N curve.

We might be able to research the relationship between LFFD and PSF if the reviewer can provide some details regarding how the PSF have been defined. The authors believe that for including the LFFD effect in design, further discussion between designers and authors would be necessary. Therefore, if there are any recommendations, we would be happy to consider them.

6) Figure 6 is unclear. How is this damage presented to be considered over the lifetime of the structure as the mean wind speed is not a constant over a day or a week

The objective of Fig. 6 is to show the origin of LFFD. So we showed that for example for m=3, 90% of the damages are from 10min cycles, and the rest of the damage is due to LFFD (that majority of them happen as daily cycles). As already answered in question 3, we did not actively use wind speed in the calculation of damage,

although by having the measured strains, we included the gradual change in the mean wind speed and direction, indirectly.

and therefore it is unclear how the lifetime of the structure can be evaluated without actually measuring the damage until failure.

We showed in Fig. 5, that these LFFD-factors converge to a fixed value and are constant for the lifetime of the wind turbine if we have enough measurement (one year). So by calculating the LFFD-factor from one year of measurement, we can apply the factor directly to the final damage without LFFD (10min-based) and have the final damage with LFFD.

Clarifications to the above are required before the article can be accepted.

We appreciate the valuable comments of the respected reviewer, and we hope that our answers are clear.

---

## Author Comment (AC2)

Dear Reviewer

We appreciate you reading and reviewing our work. We value your insightful comments and recommendations.

The authors present a very interesting contribution on the effect of low-frequency response on the fatigue damage accumulation of offshore wind turbine foundations however, I would like the following points to be addressed before the article is considered for publication:

1) In Figure 5a it is not completely clear what data is plotted. The legend provides information only about the black, blue and red points but nothing is mentioned about the yellow curve, which is not even discussed in the text. Moreover, it would make it easier if you could use a different color for line showing the average of converged values for all sensors.

In line 198 of the article it mentions: "Figure 5. a shows the trend of the LFFD-factor over time along the circumference (four sensors) of the mono-pile under consideration (T1) for single slope S-N curves with slopes of m=3, 4, 5 shown by purple, yellow, and blue lines." But as the reviewer mentions, we added the explanation for different colors in the caption of Fig. 5 (a) as well.

In Fig. 5 (b), we kept the colors of lines blue, as it is only focusing on the m=5 results and the line for the average of sensors is coming from averaging the blue lines in Figure 5. (a).

2) In line 246 the authors observe that "the share of cycles lasting more than a day is nearly absent in both FA and SS.", which is a bit contradicting with the results shown in Figure 6. According to latter, this statement is valid only for m=3 and partly for m=4, while a considerable percentage (>10%) of those cycles is present for m=5.

Thank you for your attentive read. That is the reason we used "nearly". However, for more clarity, the body of the manuscript is changed to "the share of cycles lasting more than a day is nearly absent in both FA and SS, for m=3, and insignificant for m=4, while a considerable percentage (>10%) of those cycles is present for m=5"

3) When looking at the results of Figure 6, one can draw the following conclusion: The main contribution of the low-frequency response to fatigue is due to variations in the mean wind speed, which in turn results in variations of the thrust force. The latter is certainly correlated to a few SCADA variables (power, rpm etc.) and therefore the LFFD as well. The authors have not explored at all these insights, which could further help in estimating the LFFD using SCADA data alone. Please provide these plots of the LFFD with some metric of the wind speed variation (std, Dirichlet energy, etc.)

We appreciate the feedback of the reviewer and we share a similar concern. Indeed a study of LFFD in relation to the SCADA data would be valuable, but we think that it is not a trivial study and needs deep research with proper SCADA data.

We performed some analysis on the standard deviation of wind speed and direction, as well as cycle counting the wind speed and direction signals. None of these studies showed reasonable results as there is no linear link between the current SCADA parameters and thrust load, and standard deviation lacks information about the time sequence of cycles, therefore it is not a proper metric for quantifying the LFFD effect. Although some SCADA parameters are used to calculate the thrust force, we also need parameters that are not usually publicly available such as the thrust and power coefficients.

The scope of the current paper (and research project) is to quantify the LFFD effect based on the measured strains and to show that in some cases it can be significant. However, in light of the reviewer's questions, some additional checks have been done, which are not included in the main body of the manuscript, as it is not in the scope of the paper. We hope the reviewer agrees that the current scope of the paper is well-defined and results can be considered standalone.

4) Following up on the previous comment, the contribution of the different sources of variability should be further explored and quantified. Namely, the low-frequency response, whose cycles are smaller than a day, seems to be owed mostly to wind speed variations. On the other hand, the contributions of longer cycles is owed to both wind speed and wind directions changes. These changes can be well quantified using the available

SCADA data and related to the LFFD. These are very substantial insights that the authors should explore and provide the corresponding plots.

As the reviewer mentioned, thrust load which is linked to the wind speed, can be a parameter that might be used to have a quantification of LFFD needless of the strain measurement. In the current work, the focus was on the LFFD effect derived from the strain time series. We added in the future work that a further study can be done to see in the case of having only the SCADA, how much of the LFFD effect can be covered and how SCADA can be used to get the LFFD factor.

As to reply to the reviewer, we tried to relate the thrust force to the LFFD. For that, we made a look-up table of thrust force for different wind speeds. Since for calculating the thrust force, we needed parameters such as Ct and Cp which are turbine/blade properties and usually not publicly available, we assumed that the Mtn bending moment has a direct link with the thrust force, and we used that instead of the thrust load. So we used the mean bending moments of 10-minute bending moment signals and grouped them for small wind speed bins of size 0.1 m/s. Then in each wind speed bin, the average of all the mean Mtn values was selected and put in the look-up table. Notably, for building this table, we only used the power-generating periods. This table was used to create a thrust load signal for all the measured period. So for each 10-minute wind speed, the corresponding thrust value (Average of mean Mtn in that wind speed bin) was selected. This thrust load signal changes only with the variation in the wind speed. To have the combined effect of variation in wind speed and direction, the thrust load which is always in the direction of the wind, is projected using Eq. 1[1] to a sensor's heading (260°), which is close to the prevalent wind direction.

$$M_h = M_{tn}cos(\alpha - \beta) \hspace{3cm} \text{Eq (1)}$$

Where $\alpha$ is a fixed heading and $\beta$ is the mean yaw of each 10-minute window.

The next plots show how different the thrust load and the projected thrust load signals behave with respect to the change in the wind speed and direction.
* * *
[1] Mai, Q. A., Weijtjens, W., Devriendt, C., Morato, P. G., Rigo, P., & Sørensen, J. D. (2019). Prediction of remaining fatigue life of welded joints in wind turbine support structures considering strain measurement and a joint distribution of oceanographic data. Marine Structures, 66, 307-322.

[Figure]

Figure 1 Different behavior of the thrust load and the projected thrust load signals for the change in the wind speed and direction. Black dashed lines are the direction of projected heading.

To see the effect of LFFD only from thrust load, we cycle counted the thrust load for different window lengths of daily, weekly, monthly, yearly, and the whole measurement which was three years. Figure 2 shows the share of full cycles of different period sizes. Because by using thrust load, we work with an average of 10-minute files, therefore, we lose the information about the cycles that happen within the 10-minute windows. So the shortest window that we can have is a daily window, as less than that gives only a few data points with not enough peaks and valleys to form a full cycle. We see that the majority of the low-frequency cycles happen within a day or a week.

[Figure]

Figure 2 Low-frequency cycles forming from thrust load (wind speed) and projected thrust load (wind speed and direction) variation in different period sizes.

Figure 2 shows that when we consider the combined effect of variations in wind speed and direction (projected thrust load), longer periods such as monthly and yearly will have a large contribution. This means that some rare but very large cycles only happen when both wind speed and direction are having a concurrent effect . Based on Figure 2, we can explain what we see in Figure 6 of the article. Figure 2 shows that while wind speed variation

alone (left figure), only causes almost up to week-long cycles (which we see for FA and SS in Fig 6 of the article), when we consider wind speed and direction together (right figure), cycles up to 3-year-long can appear (which we see in the sensors in Fig 6 of the article).

In light of this analysis, the main manuscript has been changed slightly in section 4.1. However, we propose to not include this analysis in the paper in full as in our opinion it does not fit with the original scope of the paper and dilutes the overall message. As mentioned there is a separate study required to accurately quantify the LFFD effect solely on SCADA data.

5) The discussion between lines 286 and 291 seems to be a bit contradictory to the findings presented in section 4.1 and the results shown in Figure 7. The sensors perpendicular to the dominant wind direction, meaning the ones closer to the SS direction, are the ones that seem to have lower LFFD factors, while the ones aligned with the dominant wind direction have higher LFFD factors.

Could you please elaborate further on the contradiction, as we did not notice any contradictions? Figure 7 demonstrates that the sensors normal to the prevalent wind direction (pink triangles) have the highest LFFD values. Also, Figure 6 shows for m=4, >30% share of low-frequency cycles for S3 and S6 (sensors in the direction of SS), while S1 and S5 which are mainly in FA direction have a 30% share of low-frequency cycles. So the results of this plot confirm the discussion in lines 286-291.

If this part of the article is confusing ("We observed that among the sensors normal and parallel to the dominant wind direction, the differences in base damage ("T") are larger than the differences in the added LFFD damage. Therefore, the LFFD-factor of sensors parallel to the wind is lower than the sensors normal to the wind."), we should emphasize that the differences in base damage are not included in the paper due to confidentiality. So Figure 6 in the article does not show this difference as the damages are normalized to their maximum value.

6) The existence of data from different turbines should be used to validate the insights from points 3) and 4) by exploring the data patterns between LFFD and SCADA for all four turbines (T1-T4).

As we mentioned earlier, finding a method to consider LFFD solely based on the SCADA needs a profound study and is out of the scope of this article. In addition, the environmental conditions of the four studied turbines are very similar as they are from two closeby farms, therefore, if we calculate LFFD from SCADA only, then the LFFD results would be very similar. For the study of LFFD based on SCADA, data should be collected from sites with very different environmental conditions to be able to have a reliable and verified methodology.

We would also like to mention that T1 and T2-T3-T4 are from two different wind farms with different turbine types and dimensions. Drawing a conclusion on the impact of SCADA conditions on LFFD between these sites would not be advisable due to their different size. A study for the turbines T2-T3-T4 is possible, but as mentioned they were subjected to near identical conditions. Alternatively one could look at a study over different years. But as mentioned we would like to keep the current discussion focused on the observations from the strain measurements.

7) A proofread would help improve the language in some parts of the manuscript.

Proofreading is conducted to enhance the quality of the article's English language.

We appreciate the respected reviewer's thoughtful remarks, and we hope our responses are acceptable.

---

## Author Response (AR2)

**Response to the editor:**

Dear Editor,

We sincerely appreciate your valuable feedback and prompt response to our paper. We thank you and the reviewers for taking the time and effort to read and review our work, and for providing us with insightful comments and ideas to improve the quality of the article.

Based on the feedback from the last reviewer, we have modified the body of the manuscript. The reply to the reviewer with details of some analysis was provided in the previous round of revision.

We hope that our revised version adequately addresses your expectations and meets the concerns of the respected reviewer.

Sincerely,

**Reply to the Editor's comments**

In the revised version of the manuscript the authors have addressed the majority of reviewer comments. There is one outstanding comment regarding the effect of environmental conditions (such as wind speed) that the authors should address in a minor revision before the paper is published.

Additional private note (visible to authors and reviewers only):

Dear authors, one of the reviewers made the following comment on the revised manuscript:

"The authors conclude that 65% of fatigue damage is directly related to LFFD when m=5. This is an interesting conclusion however, this number is conditioned to a few more variables (variability of wind speed, waves, etc), which are completely neglected by the authors. Therefore, I consider it important that the authors explore these effects in the quantification of LFFD."

Please address this comment in a minor revision.

As the reviewer demanded, we clarified in the manuscript text that there is a clear relation between the SCADA parameters and the LFFD-factor, but to our knowledge, there is no widely accepted way to find this link. In the conclusion, we emphasized that the results are site-specific and might change for other sites/turbines.

Furthermore, the manuscript is checked carefully once more to avoid any typo or missing information.